# The Anthropometric Measurement of Nasal Landmark Locations by Digital 2D Photogrammetry Using the Convolutional Neural Network

**DOI:** 10.3390/diagnostics13050891

**Published:** 2023-02-26

**Authors:** Nguyen Minh Trieu, Nguyen Truong Thinh

**Affiliations:** College of Technology and Design, University of Economics Ho Chi Minh City—UEH, Ho Chi Minh City 72516, Vietnam

**Keywords:** nasal landmarks, nasal geography, facial morphology, anthropometric measurement

## Abstract

Measuring and labeling human face landmarks are time-consuming jobs that are conducted by experts. Currently, the applications of the Convolutional Neural Network (CNN) for image segmentation and classification have made great progress. The nose is arguably one of the most attractive parts of the human face. Rhinoplasty surgery is increasingly performed in females and also in males since surgery can help to enhance patient satisfaction with the resulting perceived beautiful ratio following the neoclassical proportions. In this study, the CNN model is introduced to extract facial landmarks based on medical theories: it learns the landmarks and recognizes them based on feature extraction during training. The comparison between experiments has proved that the CNN model can detect landmarks depending on desired requirements. Anthropometric measurements are carried out by automatic measurement divided into three images with frontal, lateral, and mental views. Measurements are performed including 12 linear distances and 10 angles. The results of the study were evaluated as satisfactory with a normalized mean error (NME) of 1.05, an average error for linear measurements of 0.508 mm, and 0.498° for angle measurements. Through its results, this study proposed a low-cost automatic anthropometric measurement system with high accuracy and stability.

## 1. Introduction

According to many reports, the development of global cosmetic surgery is at a high positive level in Europe and Asia [1,2]. Facial cosmetic surgery is performed more and more not only in females but also in males [3]. Many studies have reported that most people are dissatisfied with the features of their bodies [4,5]. Depending on age, profession, gender, and residence area there are different aesthetic and beauty ideals [6,7]. Indeed, idealized facial beauty is ever-changing and achieved by structural harmony of the face. The nose, occupying the central position of the face, is the most important and impressive feature [8]. Achieving good results in rhinoplasty surgery requires many factors, including an understanding of the morphological characteristics of the nose and its correlation with organs in the face [9,10]. The measurement and evaluation of its anthropometric parameters and anatomical structure is important in order to have a more comprehensive view of the nasal morphology. In nasal anthropometry, there are not many studies on the root of the nose. Rather, the focus has been on the tip of the nose [8,11]. In plastic surgery, if the doctor changes the structural indicators of the nose without understanding the relationship between the parameters, it will result in undesirable results and post-surgery complications.

With the main goal of ensuring the beauty and safety of plastic surgery, research analyzing and modeling human faces using deep learning is developing rapidly and robustly. The surgical process should focus on safety and patient satisfaction, so the analysis of anthropometric indicators is studied to establish the relationship between facial parts [12]. Moreover, there are quite a few studies focusing on predicting age, gender, emotions, and human recognition [13,14,15]. In addition, there are many studies using deep learning in assessing the morphology of the face in some diseases [16]. There are many anthropometric methods used to determine the parameters and position of landmarks. The most popular method is the direct measurement method, which is used by many researchers in papers [17]. The basic measuring device used in anthropometric measurements is usually sliding calipers, and for measuring angles a goniometer is used. In these measurement methods, since the dimensions can be read directly it seems as though the measurer can feel more clear and more confident. However, this method takes a lot of time and the measurer needs a lot of experience to determine the landmarks on soft tissue accurately. In addition, it is difficult to use the direct measurement method to accurately measure sensitive soft tissue locations, such as the eyes, as the measurement is easily disrupted by blinking. During the measurement, the elasticity, thickness, and density of the soft tissue organization also cause errors. Another disadvantage of the direct measurement method is that the result depends on the adjustment process and the measuring force may change the result. Many studies have proposed anthropometric approaches that use 3D images by 3D scanners [18]. The price of 3D image collected systems is very high, so there are few medical facilities that can be used especially in developing countries. Therefore, a low-cost method for collecting anthropometric data is needed, and using 2D images from digital cameras is an approach that deserves attention. In the report by Seo, Y. S et al. [19], they compared reliability between 3D imaging and 2D photography, and the conclusion showed no difference between the measurements. Another method for anthropometric measurement of the human face is indirect measurement through normalized photographs. Photogrammetry is a technique that has supported anthropometric studies dating back to the 19th century. There are many studies on dimensions from images [20,21]. Researchers proposed general rules about head posture, camera placement, lighting conditions, and landmark recognition on the face as well as the described methods. The use of normalized photographs makes this method scientific and accurate. Photographs have become important and reliable research materials in anthropometric research and lecturing. With this measuring methodology, there are advantages: the easy determination of points and dimensions; simple operation and easy evaluation; and easy storing and exchanging of data information. In addition, the focus of the camera, lighting, and the psychology of the photographed person can affect the quality of the photograph, which in turn can lead to measurement and analysis errors. Most of the studies for recognizing landmarks on the face have focused on applications in life and entertainment. The recognition of landmarks with the support of deep learning models can be determined automatically, but the accuracy is not high because it is mainly applied in face reconstruction in VR models or to the recognition of key points on the face to detect emotions. In X-ray medical images, studies of cephalometry landmarks are based on random forest models in machine learning [22,23]. In these studies, the cluster and determination of landmarks were about 70–75% accurate. In addition, the use of artificial neural networks for binary classification yielded an accuracy of about 75.3% [24]. Some studies have used deep learning to determine the x and y coordinates of the waypoints presented in reference [25]. To increase the accuracy, the scientists also used the improved faster R-CNN model called CaphaNet [26,27]. Rao et al. [28] introduced an approach in orthodontics using the YOLO model to detect the face and then the active shape model (ASM) algorithm was also used to extract the landmarks. The results of their study show well-recognized landmarks with errors from 0–6 mm. However, most landmarks have errors that lead to errors in anthropometric measurements. The YOLO model was used only to detect the face, which reduces the speed of the main model. In another study, 18 landmarks were automatically retrieved using the anthropometric face model [29]. They identified the rotation angle of the face and calibrated it to the frontal angle in order to distinguish other portions of the face using the distance between the eyes as the primary parameter and then facial landmarks were extracted using an anthropometric face model. However, the study only corrects the face in the horizontal direction and is ineffective for significant vertical rotations. In addition, the model’s correctness is dependent on how well the two added eyes’ centers are identified.

This study proposes a method to determine landmarks and nose properties based on the convolutional neural network (CNN). The regression for landmarks is a highly nonlinear mapping function; each point has a corresponding nonlinear mapping function. This study proposes a method for the automatic recognition of facial landmarks. Medical experts label the facial landmarks for data that are used to train and test the extraction model. Anthropometric measurements, performed by medical experts, are applied to collect anthropometric data which are used in many different fields. This process takes a long time and depends on the experience of the performer, so the training costs and time are high. A low-cost automatic system is proposed to help collect anthropometric data quickly and objectively. The experiments confirmed in the final section show that this model achieves the required results compared to other methods.

In brief, a low-cost facial landmark location automatic system is introduced, and it is used to collect anthropometry indexes for plastic surgery using the CNN model. This paper includes four parts. Sustainability theories of anthropometric measurements, neoclassical facial proportions, and a deep model for landmark extraction and automatic measurements based on landmarks are presented in Section 2. Next, the experiments are detailed in Section 3 and the conclusions are outlined in Section 4.

## 2. Materials and Methods

### 2.1. Nasal Landmarks and Anthropometric Measurements

There are many studies on the anthropometry of the external nose and nasal base by direct or indirect measurement through normalized imaging. The facial landmarks are defined below based on the study of LG Farkas [30]. The nasion (denoted n), is the midpoint of the two segments of the nasal bone and the nasofrontal joint. In fact, for researching soft tissue, some studies also determined this point as the most concave point of the soft tissue in the nasofrontal joint along the midline [20]. The glabella (g) is the most convex point of the forehead on the midface. The maxilofrontale (mf) is the point located at the base of the nasal root, and is more medial than the medial corner of the eye and closer to the medial border of the orbital part of the frontal bone. The kyphion (k) (aka hump point) is the highest point on the bridge of the nose. This is where the nose structure is not straight and usually this point is removed to create a straight nose after surgery. The rhinion (r) is the point between the bone and cartilage on the nose. The alare (al) is the outermost point on the curve of each side of the nose. The alar curvature (ac) is the most lateral point on each ala’s curved baseline (alar groove). The labiale superius (ls) is the sagittal midline point of the upper lip. The pogonion (pg) is the most protrusive anterior sagittal midline of the chin [31]. In addition, the landmarks on the nasal base are defined in Figure 1. In this study, the anthropometric noses are focused; however, some facial landmarks are also detected to compare the index by neoclassical standards. The landmarks used in this stud are summarized in Table 1 with their name and symbol. The n of straight nose bridges in men is higher than in women, but the proportion of rough nasal bridges in men and women is the opposite. Vietnamese people have two main types of nose bridge: a straight nose bridge and a rough nose bridge. Based on the base of the nose and the line between the base of the nose and the tip of the nose, the bridge of the nose is divided into four types as follows. Firstly, a straight nose is defined by the line connecting the base of the nose and the tip of the nose coinciding with or deviating 1.0 mm from the bridge of the nose at the junction between the bone and the cartilage of the nose. Next, a concave nose is defined by a concave bridge below the line connecting the base of the nose and the top of the nose that coincides with or differs from 1.0 to 5.0 mm at the junction between the bone and cartilage of the nose. Thirdly, a broken nose is defined as having a concave bridge below the junction of the nose root and the tip of the nose more than 5.0 mm at the junction between the bone and cartilage of the nose. Finally, the nose is defined as having a convex bridge over 1.0 mm above the line connecting the base of the nose and the tip of the nose at the junction between the bone and cartilage of the nose.

Anthropometric measurements are non-invasive measurements for the collection of human anthropometric data, which are used in identification, and include ethnic identity, sex, and age. Measurements are taken to collect indicators that can be used as a basis for assessing the patient’s changes before and after surgery. In this study, 12 linear measurements are taken including the nasal root (mf–mf), nasal height (n–sn), nasal length (n–prn), nasal tip protrusion (sn–prn), nasal width (al–al), anatomical width (ac–ac), inter canthal width (en–en), ala length (ac–prn), nostrils floor width (sbal–sn), columella width (c’–c’), superior width of the columella (cw–cw), and ala thickness (al’–c’). Measurements are defined in Table 2 with the symbol from d_1_ to d_12_. Angular measurements are applied including 7 angles on the later view and 3 angles on the mental view. In rhinoplasty surgery, the kyphion angle is of interest to doctors and patients. The presence or absence of this angle depends on the configuration of the nasal bones and often the surgeries to remove it are highly complex. Moreover, anthropometric geometries are also used to predict syndromes and perform genetic screening [32,33]. The angles used in determining the nasal angles are summarized in Table 3 based on the recommendations of Lazovic et al. [34] and He et al. [35]. As mentioned, rhinoplasty is becoming more and more popular not only in women but also in men. Therefore, the standards of a perceived beautiful nose are proposed as the basis for rhinoplasty surgery. In Vietnam, eight neoclassical standards are used as references in planning rhinoplasty surgery and patient education, which are presented in Table 4. They were defined as beautiful facial proportions as suggested by Le et al. [36] and Porter et al. [37]. It includes the orbitonasal canon, orbital canon, naso-oral canon, nasofacial canon, three sections of the facial profile canon, a nose height equivalent to the ear length, a nose height of approx. 0.43 (n–gn), and a distance of the corner of the mouth to the nasal alare of equal distance to the corner mouth to the center of the pupil. These standards are used as a reference for patients and doctors. In addition to these criteria, the surgery must also pay attention to the patient’s respiratory problems and the revised horizontal deviation of the nose [38]. Anthropometric data are used in many different cases such as in pre- and post-operative evaluation to design humanoid robots [39].

### 2.2. Concept of the CNN Model for Determining the Location of Nasal Landmarks

#### 2.2.1. Data and Pre-Processing

Anthropometric measurements based on landmarks play an important role in evaluating the indices of the nose before and after surgery. Currently, landmark extraction and measurement are performed by experts, so these take a great deal of time and high cost for training staff. Moreover, the accuracy of figures depends a lot on measuring equipment and experience. This study proposed an automatic system to extract facial landmarks to collect data, which is used for patients undergoing rhinoplasty surgery. The dataset is collected from a digital camera with three views: frontal, lateral, and mental. Each participating volunteer was studied with three different images for training and testing model. A digital camera is used on an automated collecting system such as Figure 2, thus ensuring that the distance between the camera and people is constant in all of the images. This helps to ensure the accuracy of the extraction and conversion of the factors of the measurements. The automatic system includes a digital camera and a rotating mechanism to take three images for each patient. These images are pre-processed and labeled manually for the training model. Participants in this study were set up to collect the data at the university. This is a non-invasive data collection process that does not affect the volunteers’ health, only using images. For ease of handling, the images were collected with a green background. It speeds up the process of extracting facial landmarks. Moreover, to avoid the overtraining case, some data augmentation methods are applied such as rotation, scale, flip, etc.

Algorithms of deep learning are proposed to solve each specific problem in different fields, including natural language processing and computer vision. In the computer vision field, the convolutional neural network (CNN), which is a form of feed-forward neural network based on the share weight of kernels [40], is widely applied in many fields such as agriculture, industry, service, medical, etc. [41,42,43]. Extracting facial landmarks is not a new topic but it has not been widely applied in the medical field, especially to tracking anthropometric indexes of rhinoplasty surgery patients. Little research on landmark extraction has been published [44,45,46]. Notably, the names and locations of landmarks are determined differently according to the application. In this study, we designed a suitable CNN network to identify landmarks according to medical theory. Accuracy is considered an important factor in medical systems therefore in this study, which is an important impact to decide the success of the study. The basic structure of the CNN includes an input layer, convolutional layers, pooling layers, fully connected layers, and the final output layer. The process of landmark extraction is performed through the following steps.

Stage 1. Dataset Pre-processing

Images are taken by a digital camera (Canon EOS 60D) which is set up on a system with a green background and the size of the parent image as 5184 × 3456 pixels. The background does not need to be removed when using the CNN model for medical image analysis to collect the anthropometric index. Indeed, accuracy plays an important role. Therefore, to improve the metrics of the applied model, all the of the background should be removed by the threshold method because the background is a green color so this can be conducted quickly and easily. All images are converted to grayscale with optimal resized 416 × 416 images. The feature extraction for removing the noise is subject to increased bias due to the camera’s noise [47]. The data are manually labeled by experienced personnel in the medical field. Training data consists of 37 key points with coordinates of each defined landmark *L_i_*(*x_i_*,*y_i_*), where *x_i_* and *y_i_* are the coordinates of the *i*th landmarks in the *x* and *y* directions, respectively. Similarly, the *Oxz* are considered for the landmarks in the nasal base.

Stage 2. Determining the location of facial landmarks using the CNN model

The structure of the proposed model includes six convolutional layers (Conv2d), three max-pooling layers, and three fully connected layers shown in Figure 3. The facial images with three views are fed into a CNN model that has been kernel-convoluted to extract characteristics specific to each class. The convolutional kernel computed parameters are included in the feature map. The pooling layer, which is non-learnable, is used to decrease the model’s parameters, speed up the calculation, and prevent overfitting. The input of the CNN model is the gray image with the size of 416 × 416, which is convoluted with kernels to extract the features. The size of the input is large enough to improve the accuracy of the model, which means that the speed of the model is traded off. The facial landmarks are provided with a location for each key point represented by (*x*,*y*) or (*y*,*z*) coordinates, respectively, which are used to determine the anthropometric dimensions of the face. Then, the coordinates of the landmark are aggregated to appear as *L_i_*(*x_i_*,*y_i_*,*z_i_*) which is used to calculate the measurements.

After each Conv2d block, the normalization layer is applied to provide data consistency, which helps reduce the overfitting of the model [41]. The parameters of Adam optimizer and a learning rate of 0.001 are applied to the model for training. The output of the model is the landmark’s locations on the face. Anthropometric dimensions are applied to track the patient indexes before and after rhinoplasty surgery. The standards for a beautiful nose follow the neoclassical standards proposed by the authors [36,37]. This paper has great significance in medicine, especially plastic surgery. Moreover, anthropometry is also used as a reference for human robots or designing protective equipment.

#### 2.2.2. Automatic Anthropometric Measurements Based on Facial Landmarks

The nose is defined as the most attractive part of the face, and nose-related surgeries are also major challenges for doctors [8,10]. Plastic surgery standards are determined by anthropometric measurements and are used for monitoring the results of surgery or are used for patient education. Currently, anthropometric measurements are performed manually on the patient’s face or through 2D/3D images by medical staff or a semi-automatic system [17]. The accuracy of this process very much depends on the experience and knowledge of the medical staff, and the cost of training human resources is enormous. Therefore, in this study, a system for the automatic anthropometric measurement of patients is proposed. The coordinates of the landmarks are defined as *L_i_*(*x_i_*,*y_i_*,*z_i_*) including *L*(*x*,*y*), and *L*(*x*,*z*) coordinates depending on the viewing angle. Measurements are defined in terms of the Euclidean distance for key points, which is calculated by Equation (1) and only in the 2D axes for each stage.
(1)$$d_{\left(L_{i},L_{j}\right)}^{\mathrm{Oxy}}=\sqrt{(x_{j}-x_{i}{)}^{2}+(y_{j}-y_{i}{)}^{2}},$$
where *d* is the distance of the landmark *L_i_*(*x_i_*,*y_i_*) and *L_j_*(*x_j_*,*y_j_*) in the *Oxy* coordinate, similarly to the *Oxz* coordinate.

In the external morphology, the angles on the nose are used to evaluate changes, which are used to monitor patients before and after surgery, and also in in patient education. The human head has a 3D shape, so the coordinates of its landmarks must include three values (*x*,*y*,*z*). The *Oxyz* coordinate system is mounted at the glabella point that is presented in Figure 1. Considering the side view, the nose angles (1–8) are defined in Table 2, and are calculated by the angle between the two vectors from the point of the angle under consideration. Namely, the angle between three landmarks *L_i_*, *L_j_*, and *L_k_* at *L_j_* is calculated by the angle created by the two vectors $\vec{L_{j}L_{i}}$ and $\vec{L_{j}L_{k}}$. This angle (*θ_i_*) is defined by Equation (2):(2)$$\theta_{i}=\arccos\left(\frac{\vec{L_{j}L_{i}\cdot }\vec{L_{j}L_{k}}}{\left\|{\vec{L_{j}L}}_{i}\right\|\left\|\vec{L_{j}L_{k}}\right\|}\right),$$
where *θ_i_* is the *i*th angle which is defined in Table 2 and $\left\|\vec{L_{j}L_{k}}\right\|$ is a vector with a length of $\vec{L_{j}L_{k}}$**_._**

The measurements are calculated, and the next task of the system is to determine the scale to convert the measurement unit of measurements from pixels to actual size for easy use in future applications. This ratio (*D*) depends on the camera correction factor introduced in ref. [48], which is expressed by Equation (3).
(3)$$D=\alpha \sum_{i=0}^{M}k_{i}c^{2i}$$
where *α* is the ratio constant between the pixel size and the actual size; *k_i_* is the distortion coefficient of the *i*th pixel; *M* is the number of pixels; and *c* is the Euclidean distance between the coordinate of the *i*th pixel and the optical center of the image.

While collecting the images in the mental view, the head pose is requested at a certain angle for all participants. However, this is qualitative because each person’s head size is different, and the staff also do not check this perfectly, so the measurements are affected by the angle of the head. This manifests in an increase in error for figures. Letting *T* be the coefficient, it is the parameter of the angle of the head that is vertical in the mental view and is defined by Equation (4).
(4)$$T=\cos\left(\frac{\left\|d_{g-prn}^{m}\right\|}{\left\|d_{g-prn}^{f}\right\|}\right),$$
where $\left\|d_{g-prn}^{m}\right\|$ is the distance from landmark *g* to prn in mental view, $\left\|d_{g-prn}^{f}\right\|$ is the distance from landmark *g* to prn in lateral view.

Considering the coordinate *Oxyz*, in the two images taken from the side view and the frontal view the ratio between the two images is also considered a coefficient. Since the distance from the camera to the person is a very small change during the measurement process, this coefficient is usually very small. In this study, the coefficient was defined as the ratio of deviation about the unit length of three views and it was also used to determine the unit conversion ratio *D*. Linear measurements were performed on 2D images and to ensure the accuracy of the measurements on all three of the different images the coefficients are added. Actual measurements from two landmarks *L_i_* and *L_j_* (*d_ac_*) are converted according to Equation (5) if measurement is made on a frontal view image, Equation (6) if the measurement is taken on a lateral view image and Equation (7) if the measurement is performed on a mental view image.
(5)$$d_{ac}=d_{\left(Li,Lj\right)}^{Oxy}*D,$$
(6)$$d_{ac}=d_{\left(Li,Lj\right)}^{Oxy}*\frac{d_{s}}{d_{f}}*D,$$
(7)$$d_{ac}=d_{\left(Li,Lj\right)}^{Oxy}\cdot \frac{d_{m}}{d_{f}}\cdot D\cdot T,$$
where *d_f_* is the unit measure value in the frontal view, *d_s_* is the unit measure value in the side view, and *d_m_* is the unit value in the mental view.

## 3. Results and Discussion

This section describes the experiments performed to evaluate our scheme and discussion in this field. Nowadays, anthropometric data are collected manually based on the knowledge and experiences of the medical staff. This takes a deal of money and is time-consuming for the training process. The error of linear and angle measurements comes from measuring equipment and subjective factors from the operator. Studies on facial landmarks have been carried out in many studies with high accuracy; however, medical applications, especially in anthropometric measurement are limited [49]. Therefore, an automatic anthropometric measurement system is very necessary and suitable. This dataset is collected from 1000 participants of various ages wherein each person has a set of three images taken including the frontal, lateral, and mental view. The images are labeled data on the location data sheet of medical students. The matching process is also checked with the measurement of the physical dimensions of the nose with a caliper. The system is evaluated based on the comparison between 12 linear distances and 10 angles from 37 facial landmarks between the proposed system and the manual method by medical staff. Anthropometric measurements focus on the nose to collect the patient’s anthropometric indicators before and after rhinoplasty surgery. Landmarks are carefully marked by medical staff on 203 patients to evaluate the accuracy of the system. The dataset is described in Table 5 including the number of participants, sex, and age. Participants were required to be free of nasal deformities, and to perform facial anthropometric data collection by non-invasive measurement. The actual system proposed in this work is shown in Figure 4.

An anthropometric data acquisition system is proposed to ensure that the focal length of the camera and the distance from the camera to the participant are unchanged to ensure the conversion coefficients. Landmarks are defined using a CNN model with input from the digital camera and then pre-processing is applied to reduce the noise and then data are fed into the CNN model. The location of nasal landmarks is extracted with each landmark *L*(*x*,*y*) in the frontal view, *L*(*x*,*z*) in the lateral view, or *L*(*y*,*z*) in the mental view. Then, the coordinates of the landmarks are aggregated according to *Oxyz* coordinates, namely *L*(*x*,*y*,*z*) to be used for reconstruction studies from anthropometric measurements. From landmarks, the anthropometric measurements are made using the formulas presented in Section 4. Figure 5 depicts the proposed implementation for the automatic measurement of anthropometric values. Stage 0 is images collected from the camera system. In stage 1, the images are preprocessed with steps such as resizing, reducing noise, and converting gray color. Next, the image is fed into the CNN model to extract the location of the facial landmarks. Finally, calculations are applied to determine the patient’s anthropometric dimensions. These metrics are then used to monitor patients or are compared with neoclassical standards.

### 3.1. Dataset for Evaluation

The data used for the evaluation of the model’s accuracy are not used for the training process. The data include 203 participants with each person collecting images at three views: frontal, mental, lateral, and marked anthropometrically by specialized staff. The data used for this process consist of 609 images, with a male ratio of 0.49. Images are collected from a digital camera with a green background and are collected from the proposed capture system so the distance from the person to the camera is unchanged during data collection. All collected images are resized to 416 × 416 to match the model’s input. These metrics are used for comparison with the system’s automated measurements.

### 3.2. Evaluation of the Accuracy of Landmark Extraction

In this experiment, the location of ground truth and predicted landmarks are compared using the dataset of 203 participants. Thirty-seven points are extracted with each point having coordinates *x*,* y*, and *z* for each view. The error of landmarks as evaluated by normalized mean error (NME) is calculated by Formula (8). From three views, each modified and updated landmark has the form *L_i_*(*x*,*y*,*z*) with the coordinate system shown in Figure 1. In this study, occlusion cases cannot occur because this is the data collection process, participants are required to stay in fixed positions when collecting images.
(8)$$NME=\frac{1}{N}\left(\sum_{n=1}^{N}\frac{\sqrt{{\left(x_{i}-\bar{x_{i}}\right)}^{2}}}{W}+\frac{\sqrt{{\left(y_{i}-{\bar{y}}_{i}\right)}^{2}}}{L}+\frac{\sqrt{{\left(z_{i}-{\bar{z}}_{i}\right)}^{2}}}{H}\right),$$
where *x*, *y*, and *z* are the coordinates of the *i*th landmark. W, *L*, and *H* are the width, length, and height of the shapes, respectively, for each view.

All images are labeled and fed into the CNN model to perform the learning of the parameters shown in Figure 6. In each block, the learnable parameters are shared and updated after the training epoch. Figure 7a shows the results of identifying 37 landmarks on a participant. The error evaluation of the above process is performed by comparing the actual coordinates and the predicted *L_i_* coordinates. Actual coordinates are collected by qualified personnel and are considered to be absolute. The average error value in the *x*, *y*, and *z* axes is 0.530, 0.555, and 0.474 mm, respectively, and the standard deviation is 0.269, 0.295, and 0.298 mm, respectively, which means *NME* = 1.05%. This result is satisfied by experts in the field of anthropometry. To objectively assess the accuracy of the model, we compared the *NME* defined in Equation (7) and the failure rate summarized in Table 6. Failure was defined as those points with an *NME* greater than 8% [44].

The CNN model is set up including six Conv layers with a learning rate of 0.001 and an Adam optimize function. The image is fed into the model to train the recognition of 37 landmarks over a space of 416 × 416 with the gray-color space and is trained over 100 epochs to achieve the accuracy shown. The entire program performed in this study was introduced on a Tesla K80 GPU provided by Google Colab. Figure 8 shows the model loss and accuracy of the proposed model where the red line shows training accuracy and loss, and the blue line shows testing accuracy and loss, respectively, and layer parameters of the proposed CNN model are shown in Table 7. The error of landmarks is evaluated based on Formula (7) and Table 6 shows a comparison with the study of Hong et al. [44]. In fact, this comparison only shows that the NME of this model is satisfied when compared to the results of the landmark recognition contest and there is no statistical significance in comparing NME between studies because they are not evaluated on the same dataset.

### 3.3. Evaluation of the Accuracy of Anthropometric Measurements

In this experiment, 12 measurements and 9 measuring angles (for participants without hump *K*) or 10 measuring angles (for participants with hump *K*) are taken on 203 people. The data are used to compare the results of manual measurements with the automatic measurement process by the system. Anthropometric data are used to collect the patient’s readings before and after the surgery to help the doctors keep track of changes. The criteria of facial beauty are defined in Table 3 and are used to evaluate beauty according to neoclassical standards. The deviation of 12 linear measurements is evaluated using Formula (9).
(9)$$e=\left|d_{i}-\bar{d_{i}}\right|,$$
where *d_i_* is the actual measurement taken by an expert in this field; ${\bar{d}}_{i}$ is the measurement performed by the automated system.

In fact, the accuracy of the measurements depends largely on the accuracy of the coordinates of the landmark location and the conversion coefficients. The coordinate system of the automatic measuring system is mounted on the patient’s face so face angles will not affect the measurement results. In this study, the anthropometric indices of the nose were focused on for application in rhinoplasty surgery; however, for comparison according to neoclassical standards, some other facial measurements were also taken. Linear and angular measurements are shown in Figure 7b, encompassing measurements used in the clinical diagnosis and monitoring of patient indexes. Anthropometric data are used in identification, racial discrimination, and human gender prediction using the structure of human internal structures such as bones, soft tissues, etc. [50]. Alternatively, data can be used as a basis for designing protective gear for workers in different areas, especially as this can also serve as a reference for making human-shaped robots. The measurement angles are used as the basis of assessment before and after rhinoplasty surgery. Aside from the facial beauty factor, the angles at the nasal base are determined so that the patient’s airways can avoid respiratory problems. Seven angles are identified in the lateral view for patients with a hump *K* that are marked from *g1* to *g7* in Figure 7b and three angles in the mental view assess the patient’s airway.

Linear distances are evaluated for accuracy through a dataset consisting of 203 participants, both male and female, and this dataset is different from the training dataset. The distance error is shown in Figure 9a, with the average error of the distances being 0.508 mm. The error values are accepted in the field of anthropometric collection. Figure 9b shows the deviation of the measuring angles which are extracted at the lateral view, which is defined as the deviation between the actual measured angle value and the measured values predicted. The mean error of all 10 measurements is 0.498° and the mean deviation is 0.287°. As for *g7*, this angle exists only when there is a hump point k on the participant’s nose, which is required to approximate 180° after corrective surgery. Anthropometric data of the measured angles are recorded to evaluate the performance of the surgical process. The values representing the anthropometric dimensions of a beautiful nose are shown in Table 4 and are used as a reference for rhinoplasty surgeries. This study is meaningful in collecting anthropometric data for regions and countries in medicine. Moreover, these data are used as references in studies on the characteristics of patients, properties, and gender.

## 4. Conclusions

In this study, a framework is proposed to automatically locate facial landmarks based on three 2D images from three views. Moreover, an automatic system is introduced to collect the image from patients. For medical image analysis, landmarks are defined according to the sustainability theories of medicine, and the accuracy of the recognition processes is highly valued in this field. The landmarks are detected in order to define the facial morphology and appearance, disease-specific geometric characteristics, and local texture features. Anthropometric data are used in many different fields: they are used as a reference for clinical diagnosis for surgery, especially rhinoplasty, and they provide a basic theory for surgery to create a beautiful nose according to perceived standards. In addition, it is used as an input parameter for humanoid robot studies. Finally, the automatic landmark extraction system is satisfied with an NME of 1.05 when tested on 203 Materials. And no file in redmine, please confirm if this part should be deletedlocal participants. Anthropometric measurements were carried out and achieved high accuracy, with an average error for linear measurements of 0.508 mm and 0.498° for angle measurements. Automatic anthropometric measurement, with the help of computers, is a low-cost method which reduces a great deal of pressure on medical staff.

## Figures and Tables

**Figure 1 diagnostics-13-00891-f001:**
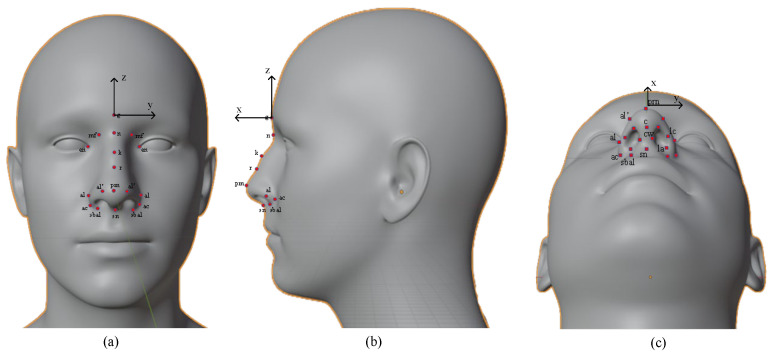
Location of the facial landmarks in three views. (**a**) Frontal view. (**b**) Lateral view. (**c**) Mental view.

**Figure 2 diagnostics-13-00891-f002:**
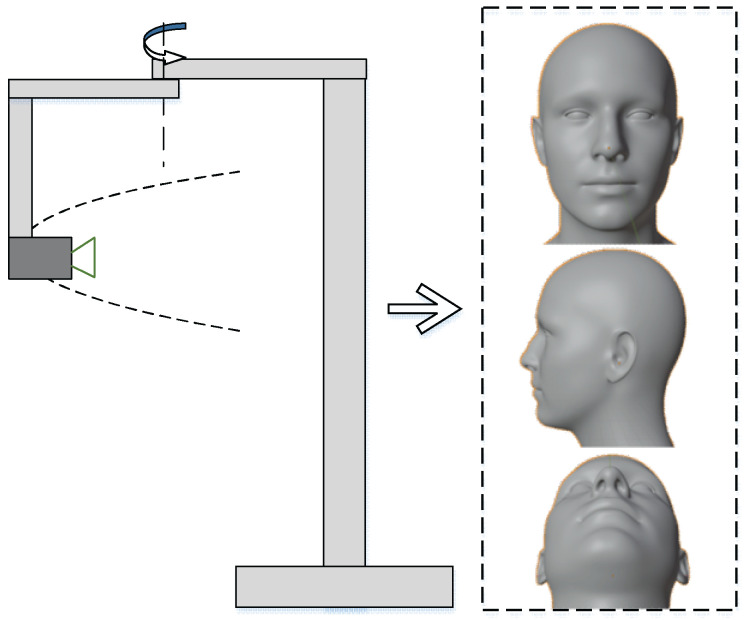
An automatic system is proposed to collect the data (including one digital camera and one rotating mechanism).

**Figure 3 diagnostics-13-00891-f003:**
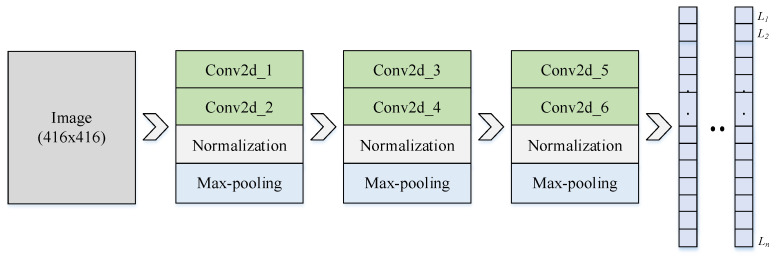
The structure of the proposed model to extract the landmarks, including six conv2d layers, three pooling layers, and three fully connected layers.

**Figure 4 diagnostics-13-00891-f004:**
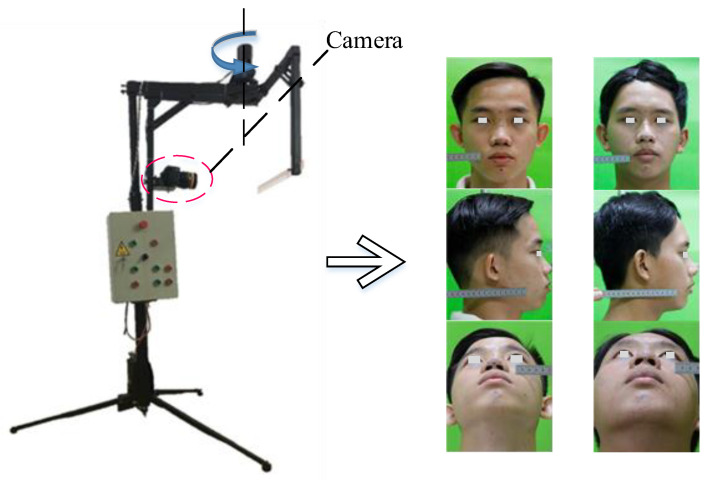
An actual system to collect the data.

**Figure 5 diagnostics-13-00891-f005:**
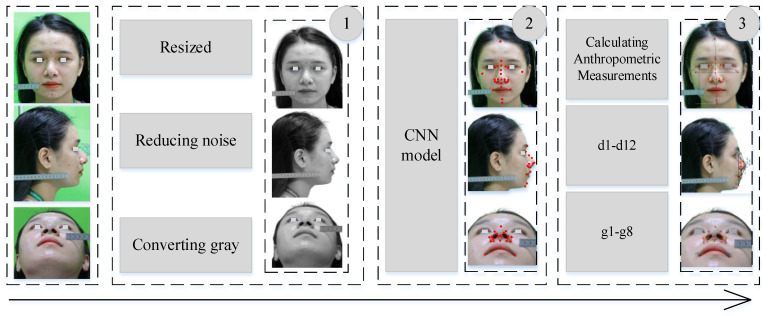
Illustration of a proposed anthropometric measurement process including three stages using the CNN model combined with anthropometric measurements.

**Figure 6 diagnostics-13-00891-f006:**
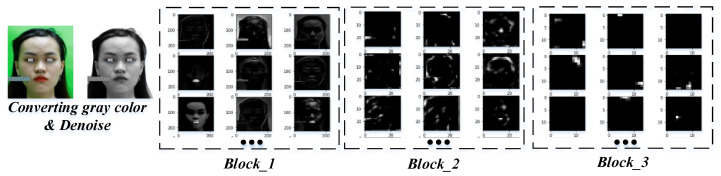
Visualizing the feature map of each block in the CNN model.

**Figure 7 diagnostics-13-00891-f007:**
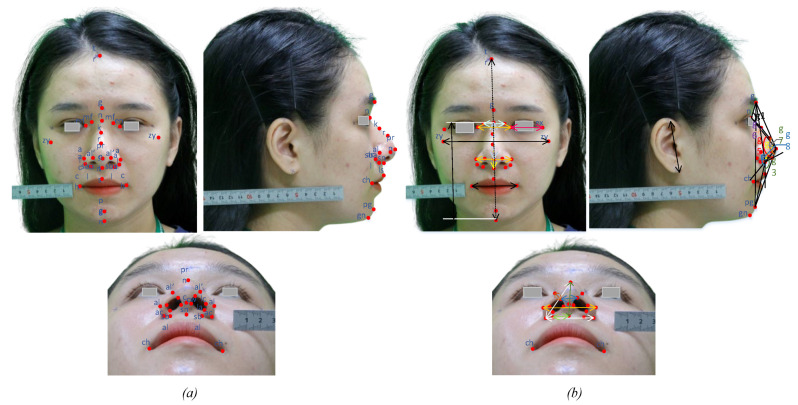
(**a**) A total of 37 landmarks are extracted from 3 views by the proposed system. (**b**) Measurements are made by the proposed formulas.

**Figure 8 diagnostics-13-00891-f008:**
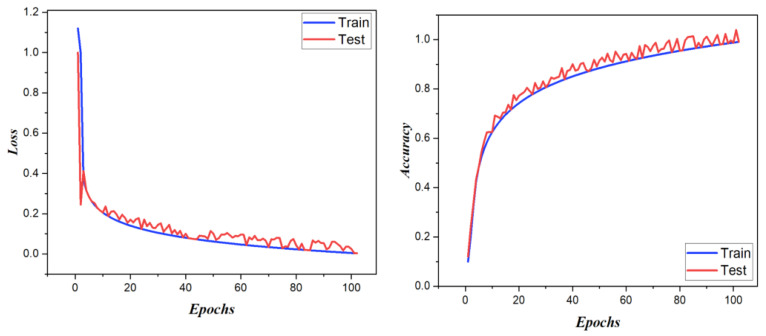
Training loss and testing loss and training accuracy and testing accuracy for the proposed model.

**Figure 9 diagnostics-13-00891-f009:**
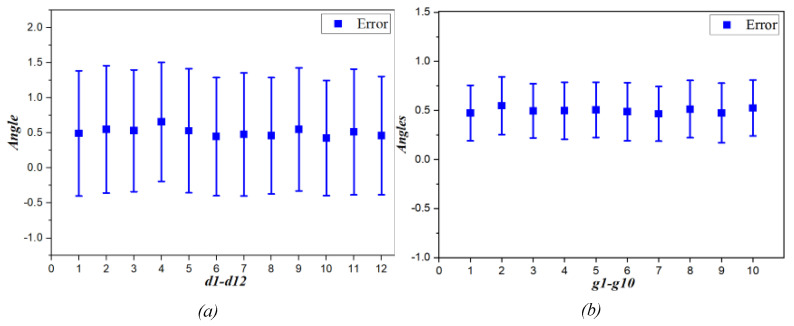
Error chart of measurements. (**a**) The error of the 12 linear measurements. (**b**) The error of the angle measurements.

**Table 1 diagnostics-13-00891-t001:** Names and symbols of facial landmarks that are used in this study [20,30,31].

Number	Landmark	Symbol	Number	Landmark	Symbol
1	Trichion	tr	13	Labiale superius	ls
2	Glabella	g	14	Pogonion	pg
3	Nasion	n	15	Zygion	zy
4	Endocanthion	en	16	Maxillofrontale	mf
5	Exocanthion	ex	17	Subalare	sbal
6	Pronasale	prn	18	Columellar peak	c
7	Kyphion	k	19	Columellar waist	cw
8	Rhinion	r	20	Lateral crus	lc
9	Subnasale	sn	21	Lateral alar	la
10	Alare	al	22	Soft triangle	c’
11	Alare’	al’	23	Cheilion	ch
12	Alar curvature	ac			

**Table 2 diagnostics-13-00891-t002:** Defining linear anthropometric measurements.

Symbol	Measurements	Distance
d_1_	Nasal root	mf–mf
d_2_	Nasal height	n–sn
d_3_	Nasal length	n–prn
d_4_	Nasal tip protrusion	sn–prn
d_5_	Nasal width	al–al
d_6_	Anatomical width	ac–ac
d_7_	Inter canthal width	en–en
d_8_	Ala length	ac–prn
d_9_	Nostril floor width	sbal–sn
d_10_	Columella width	c’–c’
d_11_	Superior width of the columella	cw–cw
d_12_	Ala thickness	al’–c’

**Table 3 diagnostics-13-00891-t003:** Anthropometric angles.

Symbol		Name	Angle
g1	Lateral view	Nasofrontal	g–n–prn
g2		Nasomental	n–prn–pg
g3		Facial convexity	g–sn–pg
g4		Nasal tip	n–prn–sn
g5		Nasolabial	c–sn–ls
g6		Nasofacial	n–prn and g–pg
g7		Kyphion	n–k–r
g8	Mental view	Alar slope	al–prn–al
g9		Interaxial	nostril axis-nostril axis
g10		Nostril axis	nostril axis-horizontal plane

**Table 4 diagnostics-13-00891-t004:** Neoclassical facial proportions.

	Standard	Symbol
1	Orbitonasal Canon	en–en = al–al
2	Orbital Canon	en–en = ex–en
3	Naso-oral Canon	ch–ch = 1.5 (al–al)
4	Nasofacial Canon	al–al = 0.25 (zy–zy)
5	Threesection Facial Profile Canon	n–sn = 1/3 (tr–gn)
6	Nose Height Equal to Ear Length	n–sn = sa–sba
7	Nose height approx. 0.43 (n–gn)	n–sn = 0.43(n–gn)
8	Distance of the corner of the mouth to nasal alare of equal distance to the corner of the mouth to the center of the pupil	ch–en = ch–center (pupil) (Horizontally)

**Table 5 diagnostics-13-00891-t005:** Characteristics and standard parameters of datasets.

Characteristic	Training Dataset	Evaluation Dataset
Number of participants	n = 1000	n = 203
Number of Images	3000	609
Male	152	78
Female	848	125
Age	28.09 ± 12.32	23.09 ± 12.32

**Table 6 diagnostics-13-00891-t006:** Comparison of NME and the failure rate between the proposed method and the method proposed in the study by Z. Hong et al.

	Failure Rate (%)	NME (%)
Z. Hong et al. [44]	0.1	1.31
Our	0.00	1.05

**Table 7 diagnostics-13-00891-t007:** Layer parameters of the CNN model.

Layers	Filter Numbers	Filter Size	Stride
Conv2d_1	32	5 × 5	1
Conv2d_2	32	3 × 3	1
Conv2d_3	64	3 × 3	1
Conv2d_4	64	3 × 3	1
Conv2d_5	128	3 × 3	1
Conv2d_6	256	3 × 3	1
All pooling layer	-	2 × 2	2

## Data Availability

Not available.

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
