# Peer review of "The Anthropometric Measurement of Nasal Landmark Locations by Digital 2D Photogrammetry Using the Convolutional Neural Network"

_diagnostics, 2023, doi:10.3390/diagnostics13050891_

Round 1

Reviewer 1 Report

Summary:  In this study, the CNN model is introduced to extract facial landmarks based on medical theories, which learns the landmarks and recognizes them based on feature extraction during training. The comparison between experiments has proved that the CNN model can detect landmarks depending on desired requirements. Anthropometric measurements are carried out by automatic measurement divided into three images with frontal, lateral, and mental views. Measurements are performed including linear distances and angles. The results of the study were evaluated as satisfactory with a normalized mean error (NME) of 1.05, an average error for linear measurements of 0.508 mm, and 0.498° for angle measurements. Through its results, this study proposed a low-cost anthropometric measurement system automatically with high accuracy and stability.

I recommend the following revision for this work: 

1) "Depending on age, profession, gender, and residence area, there are different levels of aesthetics or beauty."- add reference for this statement. 

2) "Achieving good results in rhinoplasty-plastic surgery requires many factors, in which, it is necessary to understand the morphological characteristics of the nose and the correlation of the organs in the face." - add reference for this statement. 

3) "Currently, research on analyzing and modeling human faces using deep learning is developing rapidly and strongly. There are quite a few studies focusing on predicting age, "- add justification for this statement. 

4) "For measuring angles, a multi-purpose facial angle meter is used. In these measurement methods, the dimensions can be read directly, it sounds like the measurer feel more clear and more confident. However, this method takes a lot of time to measure, and the measurer needs a lot of experience to determine 53 landmarks on soft tissue accurately."- add justification

5) Add the clear problem statement under the introduction section. 

6) What are the major contributions of this work? add under the introduction section. 

7) The related work should be summarized in cutting edges and gaps

8) Add reference for table 1. 

9) "Algorithms of deep learning are proposed to solve each specific problem in different fields including natural language processing, and computer vision. In the computer vision field, the Convolutional neural network (CNN),"- add justification. 

Reviewer 2 Report

I have several questions:

1.- Additional parameters should be provided in order to be able to check the robustness of the system.

2.- Has any validation method been used?

3.-Has any subset of data completely independent of the training data been used to verify the results?

Reviewer 3 Report

This paper escribed a CNN-based automatic landmark detection from facial photographs.

The design of CNN is good, but there are some problems for publication.

Followings are the list to be revised.

Overall, the manuscript is redundant. There are many repetitions in text. The authors should revise the text to be simpler. I recommend that the manuscript should be proofread in English.

No description regarding an ethical approval of this study is found. The authors must state it clearly.

Statistics of participants are not clear (age distribution, sex). The authors should show them in table format.

In figure 8, the loss and accuracy has not converged. I am wondering why the authors don’t increase epochs to obtain better results.

Round 2

Reviewer 1 Report

No more comments. Authors well revised this version.